# Mobile Robot Indoor Positioning Based on a Combination of Visual and Inertial Sensors

**DOI:** 10.3390/s19081773

**Published:** 2019-04-13

**Authors:** Mingjing Gao, Min Yu, Hang Guo, Yuan Xu

**Affiliations:** 1Institute of Space Science and Technology, Nanchang University, Nanchang 330031, China; 416118717053@email.edu.ncu.cn; 2College of Computer Information and Engineering, Jiangxi Normal University, Nanchang 330022, China; myu@jxnu.edu.cn; 3School of Electrical Engineering, University of Jinan, Jinan 250022, China; cse_xuy@ujn.edu.cn; 4School of Control Science and Engineering, Shandong University, Jinan 250100, China

**Keywords:** robot positioning, visual sensor, inertial sensor, SIFT algorithm, adaptive fade-out extended Kalman filter

## Abstract

Multi-sensor integrated navigation technology has been applied to the indoor navigation and positioning of robots. For the problems of a low navigation accuracy and error accumulation, for mobile robots with a single sensor, an indoor mobile robot positioning method based on a visual and inertial sensor combination is presented in this paper. First, the visual sensor (Kinect) is used to obtain the color image and the depth image, and feature matching is performed by the improved scale-invariant feature transform (SIFT) algorithm. Then, the absolute orientation algorithm is used to calculate the rotation matrix and translation vector of a robot in two consecutive frames of images. An inertial measurement unit (IMU) has the advantages of high frequency updating and rapid, accurate positioning, and can compensate for the Kinect speed and lack of precision. Three-dimensional data, such as acceleration, angular velocity, magnetic field strength, and temperature data, can be obtained in real-time with an IMU. The data obtained by the visual sensor is loosely combined with that obtained by the IMU, that is, the differences in the positions and attitudes of the two sensor outputs are optimally combined by the adaptive fade-out extended Kalman filter to estimate the errors. Finally, several experiments show that this method can significantly improve the accuracy of the indoor positioning of the mobile robots based on the visual and inertial sensors.

## 1. Introduction

With the continuous expansion of the robot field, service robots have begun to appear in recent years, mainly engaged in maintenance, repair, transportation, cleaning, security, and other work. The prerequisite for realizing the use of these robots is to provide them with the current surrounding three-dimensional information, and, at the same time, be able to accurately determine their position. However, traditional GPS positioning technology is limited due to the diversity and uncertainty of the working environment of the robots, for example, using GPS, it is difficult to locate among very high buildings, deep underwater, or indoors. In order to realize both mapping and self-positioning, the robots carrying several sensors have to obtain the surrounding information, and establish the environment model without prior information while in motion; at the same time, they have to estimate their own motion trajectories (simultaneous localization and mapping, SLAM) [1].

In the past, a laser radar was often used as a sensor to collect the surrounding information and positioning, that is, laser SLAM, but the price of the laser radar is high. In recent years, visual SLAM using cameras as sensors to collect the surrounding information has become a research hotspot [2]. The current visual sensors applied to SLAM now include Kinect and Xtion. Kinect has limitations in speed and accuracy when it is used as a visual sensor to collect the surrounding data [3,4,5,6]. In terms of precision, Kinect is sensitive to the amount of illumination. When the lighting conditions are not good, feature point extraction and the matching algorithm will fail [7]. In terms of speed, the corresponding image processing algorithms are complex and the parallelism is low due to the large amount of image data, resulting in a processing speed that is too slow to meet the real-time requirements. In contrast, IMU has a high-frequency update and performance in real-time. It can overcome the speed limitations of Kinect, but has the problem of cumulative errors. Therefore, it is necessary to combine the two for indoor positioning.

In the field of visual and IMU integrated navigation, many experts have obtained some important results. The concept of aiding Inertial Navigation with Vision-Base SLAM to compensate for Inertial Navigation divergence was presented in [8], which clearly showed that an integrated navigation system enables better positioning and navigation of the vehicles.

In this paper, combined indoor positioning is divided into three steps. First, the Kinect depth camera is used for positioning indoors, during which the continuous color image and depth image under the multi-frame are acquired by the Kinect, the improved SIFT algorithm is used for matching the features, and the graphics processing unit (GPU) is used to improve the calculation efficiency. The random sample consensus (RANSAC) algorithm is used to eliminate the mismatching, in order to improve the positioning accuracy, and the absolute orientation algorithm is then used to obtain the pose and the motion of the robot, and iteratively obtain the motion trajectory of the robot in space. Second, the data of both Kinect and IMU are loosely combined by Kalman filtering to improve the positioning accuracy. Finally, several experiments on the visual sensor with the IMU are performed.

## 2. Kinect Positioning

### 2.1. Principle of Kinect Camera

The Kinect depth camera uses a speckle-based source calibration technique that projects a random lattice, and then captures the lattice with the complementary metal-oxide-semiconductor (CMOS) sensors [9]. When the depth information of the space changes, the dot information captured by the Kinect camera will change accordingly, and the depth information can be calculated. The Kinect camera provides three main types of raw data, including deep data flow, color data flow, and raw audio streams. In this paper, the deep data stream and color data stream collected by Kinect were mainly used to realize the perception of robot displacement and offset.

### 2.2. Getting Data

Assuming that the pixel coordinates of a feature point are (*x*, *y*), the three-dimensional coordinates of the point in the Kinect coordinate system can be obtained from the depth image by the pinhole camera model (1) after the distortion correction:(1)$$\begin{array}{c} x_{c}=\frac{\left(x-u_{0}\right)z}{f_{x}}\\ y_{c}=\frac{\left(y-v_{0}\right)z}{f_{y}}\\ z_{c}=z \end{array}$$
where z represents the depth values of the feature points, and $u_{0}$, $v_{0}$, $f_{x}$, and $f_{y}$ are the internal parameters of the red, green, blue (*RGB*) camera, where $f_{x}$ and $f_{y}$ represent the focal lengths in *x, y* directions, respectively, and $u_{0}$ and $v_{0}$ are the coordinates of the main point. The Kinect coordinate system is shown in Figure 1.

### 2.3. SIFT Algorithm

The SIFT algorithm was proposed by Professor Lowe at Columbia University [10]. It is based on the idea of image feature scale selection. Firstly, the image Gaussian difference space is established, and the extreme points in the whole scale space are detected. After filtering the extreme points with a poor quality, the sub-pixel position can be accurately determined. At the same time, the scale value is obtained to improve the resistance of the feature points to the scale change. Finally, the feature descriptor is extracted. The SIFT algorithm is computationally complex, with each feature point containing 128 feature description vectors. The SIFT algorithm has the disadvantage of tracking the targets in real-time [11,12].

#### 2.3.1. Establishment of Scale Space

The definition of the scale space is as follows [13]:(2)L(x,y,σ)=G(x,y,σ)×I(x,y)
where *L* consists of the coordinates (*x, y*) of each key point and σ, which is the scale value. Here, “*” represents a convolution operation, I(x,y) is the original image, and G(x,y,σ) represents Gaussian blur, which is defined as follows: (3)$$G(x,y,\sigma )=\frac{1}{2\pi \sigma^{2}}e^{-\frac{{\left(x-\frac{m}{2}\right)}^{2}+{\left(y-\frac{n}{2}\right)}^{2}}{2\sigma^{2}}}$$
where *m* and *n* represent the dimension of the template of Gaussian blur, (*x*, *y*) represents the position of the pixel in the image, and σ represents the scale value. The larger the scale value, the more contour features the image has. The smaller the scale value, the more detailed features the image has.

The Gaussian difference scale space (DOG, Difference of Gaussian) is constructed by using Equation (4): (4)D(x,y,σ)⋅I(x,y)=[G(x,y,kσ)−G(x,y,σ)]⋅I(x,y)=L(x,y,kσ)−L(x,y,σ)
where *k* is a constant.

#### 2.3.2. Positioning Feature Point

In the scale space, a comparison of 26 points between the intermediate layer detection point and the eight adjacent points of the same scale and the 9 × 2 points corresponding to the upper and lower layer adjacent scales is performed [14]. If the point is the maximum or minimum value, the point is an extreme value on the image as a candidate feature point. Further screening work is performed after all of the candidate feature points are extracted, including noise removal and the edge effect. A set of points to be selected is fitted to a three-dimensional quadratic function to remove the low-contrast points, and the edge effect is then determined according to the size of the principal curvature of the candidate to be selected.

#### 2.3.3. Determine the Key Point Direction

The direction value of each key point is specified by the gradient direction distribution characteristics of the key pixels of the neighborhood, so that the operator has rotation invariance. The gradient size and direction at the position are [13,14]:(5)$$m\left(x,y\right)=\sqrt{{\left(L\left(x+1,y\right)-L\left(x-1,y\right)\right)}^{2}+{\left(L\left(x,y+1\right)-L\left(x,y-1\right)\right)}^{2}}$$
(6)$$\theta (x,y)={tan}^{-1}(\left(L\left(x,y+1\right)-L\left(x,y-1\right)\right)/\left(L\left(x+1,y\right)-L\left(x-1,y\right)\right))$$
where *L* represents the coordinates (*x, y*) of the key point without the scale value σ.

The above are the modulus equation and direction equation of the gradient at (*x*, *y*). 

#### 2.3.4. Feature Descriptors

Each SIFT feature point contains information, such as the position, scale, direction, and 128 feature vectors, which removes the influence of factors such as image scale change and angle rotation.

#### 2.3.5. Color Invariants 

The color invariant model is based on Kubelka–Munk’s theory [15] and is used to describe the spectral radiation characteristics of an object: (7)$$E(\lambda ,\vec{X})=e\left(\lambda ,\vec{x}\right){\left(1-\rho_{f}\left(\vec{x}\right)\right)}^{2}R_{\infty }\left(\lambda ,\vec{x}\right)+e\left(\lambda ,\vec{x}\right)\rho_{f}\left(\vec{x}\right)$$
where λ represents the wavelength, $\vec{x}$ is a two-dimensional vector, $e(\lambda ,\vec{x})$ represents the spectral intensity, $\rho_{f}(\vec{x})$ is the Fresnel reflection coefficient at x, $R_{\infty }(\lambda ,\vec{x})$ is the reflectivity of the material, and $E(\lambda ,\vec{x})$ represents the spectral reflection of the observer’s viewing direction. All non-transparent material objects can be described by a sub-model.

### 2.4. Improved SIFT Algorithm

#### 2.4.1. Image Color Invariant Preprocessing

The classic SIFT algorithm converts a color image directly into a grayscale image. In this paper, the color invariant coefficient, *H* (*x*, *y*), of the color image is first determined, and the grayscale value is used instead. Under the standard CIE-1964-XYZ of the human visual system [16], the approximate relationship between the pixel points *RGB* and $E,\;E_{\lambda },\;E_{\lambda \lambda }$ can be obtained as follows:(8)$$\left[\begin{array}{c} E\\ E_{\lambda }\\ E_{\lambda \lambda } \end{array}\right]=\left[\begin{array}{ccc} \begin{array}{c} 0.06\\ 0.30\\ 0.34 \end{array} & \begin{array}{c} 0.63\\ 0.04\\ -0.60 \end{array} & \begin{array}{c} 0.27\\ -0.35\\ 0.17 \end{array} \end{array}\right]\cdot \left[\begin{array}{c} R\\ G\\ B \end{array}\right]$$
where *E* represents the intensity information of the image, $E_{\lambda }$ represents the blue-yellow channel, and $E_{\lambda \lambda }$ represents the green-red channel information. The data of the three channels of the color image (the original image *RBG*) is substituted into (8), and the color invariant coefficient *H* is obtained, instead of the grayscale image as the input image.

#### 2.4.2. Improvement of SIFT Feature Descriptors

The classical SIFT algorithm assigns a principal direction to each feature point in order to ensure the rotation invariance of the descriptor [17]. If the descriptor itself has an anti-rotation capability, then the rotation of the entire description area can be omitted. The circle itself has a good rotation invariance, and there are many advantages to the original square area [18]. Therefore, the ring is used to construct the descriptor of the SIFT feature point. The new descriptor is structured as follows:

Centering on the feature points, we select a circular area with a radius of 8, and divide the circle into the concentric circles of the different radii represented by different colors in Figure 2. The pixels in the same ring only change after the image is rotated, and the relative position information with respect to other pixels does not change. Similar to the previous SIFT algorithm, the mode and direction of the gradient of each feature point are calculated first, and the gradient accumulation values of the eight directions are calculated and counted in each ring. Finally, the calculated gradient values are sorted from large to small.

Gaussian weighting is performed for the entire circular area to reduce the influence of pixels away from the feature points on the feature point gradient information. The farther the pixel point $f_{i}(x,y)$ is from the central feature point $f_{i}$, the smaller the amount of information contributed to the descriptor, and the weighting coefficient is
(9)$$w_{i}\left(x,y\right)=exp\{-[{\left(x-x_{i}\right)}^{2}+\left(y-y_{i}{)}^{2}\right]/2a^{2}\}/2\pi a^{2}$$
where $\left(x_{i},y_{i}\right)$ is the position of the central feature point, and a is the constant. The first ring of eight gradients is added as the first eight elements of the feature vector, the second ring is used as the last eight elements of the feature vector, and so on. The feature description area has a total of 8 × 8 elements, i.e., 64 accumulated values, to determine the descriptor of the feature point. The new descriptor thus produced has an anti-rotation capability and does not need to rotate the description area. Moreover, the original 128-dimensional vector of the feature descriptor is reduced to a 64-dimensional vector (64 bits), which further reduces the amount of calculation, and also saves time for the matching of subsequent feature points.

### 2.5. Absolute Orientation Algorithm

In order to improve the accuracy of the absolute orientation of the digital close-range image, Nanshan Zheng described an absolute orientation method for the close-range images based on the adjustment model under the successive correlative condition. Assuming that the model is a rigid body, the absolute orientation of the model is a spatial similarity transformation problem, which mainly contains three contents [19]:(1)Rotation of the model coordinate system relative to the target coordinate system;(2)Translation of the model coordinate system relative to the target coordinate system;(3)Determination of the scale factor of the model scaling.

The absolute orientation basic relationship is shown in (10):(10)$$\left(\begin{array}{c} X\\ Y\\ Z \end{array}\right)=\lambda \left(\begin{array}{ccc} \begin{array}{c} a_{1}\\ a_{2}\\ a_{3} \end{array} & \begin{array}{c} b_{1}\\ b_{2}\\ b_{3} \end{array} & \begin{array}{c} c_{1}\\ c_{2}\\ c_{3} \end{array} \end{array}\right)\cdot \left(\begin{array}{c} U\\ V\\ W \end{array}\right)+\left(\begin{array}{c} \Delta X\\ \Delta Y\\ \Delta Z \end{array}\right)$$
where (*U*, *V*, *W*) represents the model coordinates of the point; (X,Y,Z) represents the corresponding target coordinates; *a_i_, b_i_, c_i_* (*i* = 1,2,3) are the elements related to the rotation angles; (ΔX,ΔY,ΔZ) represents the translation of the coordinate origin; and *λ* is the scale factor. 

Suppose $M\left(X_{M},\;Y_{M},\;Z_{M}\right)$, $N\left(X_{N},\;Y_{N},Z_{N}\right)$, $P\left(X_{P},\;Y_{P},Z_{P}\right)$, and $Q\left(X_{Q},\;Y_{Q},Z_{Q}\right)$ are the positions under the initial coordinates of Kinect, and $m\left(U_{m},V_{m},W_{m}\right)$, $n\left(U_{n},V_{n},W_{n}\right),p\left(U_{p},V_{p},W_{p}\right),$ and $q\left(U_{q},V_{q},W_{q}\right)$ are the coordinates under the adjacent initial position.

The coordinates of the above four points *M, N, P,* and *N* are substituted into Equation (10), and the equations of the point *M* are subtracted from the equations of the points *N, P,* and *Q*, respectively. Then, one can eliminate the translation parameters. If we suppose λ is equal to 1, the result is
(11)$$\left(\begin{array}{c} X\\ Y\\ Z \end{array}\right)=\lambda \left(\begin{array}{ccc} \begin{array}{c} a_{1}\\ a_{2}\\ a_{3} \end{array} & \begin{array}{c} b_{1}\\ b_{2}\\ b_{3} \end{array} & \begin{array}{c} c_{1}\\ c_{2}\\ c_{3} \end{array} \end{array}\right)\cdot \left(\begin{array}{c} U\\ V\\ W \end{array}\right)+\left(\begin{array}{c} \Delta X\\ \Delta Y\\ \Delta Z \end{array}\right)$$
where
$$R_{0}=\left[\begin{array}{ccc} \begin{array}{c} a_{1}^{0}\\ a_{2}^{0}\\ a_{3}^{0} \end{array} & \begin{array}{c} b_{1}^{0}\\ b_{2}^{0}\\ b_{3}^{0} \end{array} & \begin{array}{c} c_{1}^{0}\\ c_{2}^{0}\\ c_{3}^{0} \end{array} \end{array}\right]$$

The superscript zero means the initial values. The three rotation angles’ initial values of the absolute orientation (superscript zero) are
$$\Phi =-\arctan\left(a_{3}^{0}/c_{3}^{0}\right)\Omega =-\arcsin\left(b_{3}^{0}\right)K=-\arctan\left(b_{1}^{0}/b_{2}^{0}\right)$$
where Φ,Ω,K are the rotation angles.

## 3. Kinect-Based Robot Self-Positioning Indoors

The above section mainly describes the improved SIFT feature extraction and matching algorithm, the removal error matching algorithm of the random sampling consistency (RANSAC) [20], and the absolute orientation algorithm. This section will introduce the specific implementation of the positioning algorithm, which is mainly divided into three parts.

(1) Extracting the key point

First, the common key points are extracted from the images (Image1 and Image2) by the improved SIFT method, and the image point coordinates (SIFTData1 and SIFTData2) in the RGB image are respectively obtained. Then, the depth coordinates (Depth1 and Depth2) of the key points are acquired from the depth image.

(2) Reducing the error

Due to the external conditions, such as noise, light, etc., there are still many mismatching points after extracting the feature points of the image by using the improved SIFT algorithm. It is therefore necessary to further improve the accuracy of the feature points. The random sampling consistency (RANSAC) algorithm is used to eliminate the mismatched points in the matching pairs to obtain the position information (Data1, Data2). Because the camera is composed of an infrared camera, bad weather affects the distance of infrared radiation, which can be too high in intensity and saturates the sensor, thus decreasing the positioning accuracy. In order to improve the accuracy of the data, this paper uses the bubble sorting method, which selects the four key points with larger spacing and then takes the average of the three-dimensional coordinates of the nearby points around these four points as the correct results.

(3) Calculating the external parameters to obtain the trajectory

After using the improved SIFT algorithm and the RANSAC algorithm, the three-dimensional coordinates of the feature points captured by the Kinect camera at the first position are marked as Data1, and the three-dimensional coordinates of the feature points obtained at the second position are Data2; then, Data1, Data2, Depth1, and Depth2 are processed by the bubble sorting method and averaged. Following this, the absolute orientation algorithm is used to calculate the rotation matrix from which the orientation (the three directions) is obtained, and the offset values between two positions are calculated to obtain the distance between the two points. The robot is initially located at the coordinate origin. When the robot moves further to the third position through the first and the second positions, the new feature point is obtained as a new Data2, and the feature point acquired at the second position is replaced as a new Data1. After the data are updated, the relative motion parameters of the robot from the second point to the third point are calculated by the new data1 and data2, so that the trajectory of the mobile robot in space motion can be obtained by the successive iterations. The visual positioning flow is shown in Figure 3.

## 4. Strapdown Inertial Navigation Principle and Algorithm Design

The strapdown inertial navigation algorithm [21,22] uses a mathematical platform, and its core part is the attitude update solution [23]. Due to the superiority of the quaternion algorithm, a quaternion-based rotation vector algorithm is used here. 

The basic idea of the strapdown inertial navigation update is to use the navigation parameters (attitudes, speeds, and positions) of the previous moment as the initial values, and use the outputs of the inertial device (angular velocity and acceleration) to solve the navigation parameters of the current moment, that is, from the initial values to the destination. The current navigation parameters are used as the known values for the solution for the next time [24,25]. 

## 5. Adaptive Fading Extended Kalman Filter

Similar to the extended Kalman filter (EKF) method, the adaptive fading extended Kalman filter (AFEKF) method is also divided into three parts: namely, prediction, update, and map expansion. The latter differs from the former, that is, the method used for the update is different. AFEKF needs to calculate the fading factor based on the information on the current observations and the historical observations, and adjust the a priori estimates so that the system can better deal with the influence of noise interference and other factors [26].

## 6. Kinect and IMU Combined Positioning

The article [8] conducted a theoretical work and produced simulation results for the inertial navigation and vision combined system in order to benefit each other. Kinect can compensate for the inertial navigation divergence. However, the amount of visual image data is too large for a personal computer to complete the positioning calculation quickly, and the real-time navigation performance is impossible. In addition, it is limited by the light conditions as the visual positioning cannot work in a dark environment. The long-term work will also generate cumulative error, so it is necessary to introduce other navigation systems to set up an integrated system. Therefore, this paper adopts the IMU/Kinect-based Kalman filter loosely integrated navigation system to reduce the errors caused by the vision system, so that the accuracy of the mobile robots’ positioning and the autonomous navigation indoors can be improved.

Kalman filtering is a linear recursive algorithm. It is necessary to first determine the initial values in order to estimate the next moment state values with the current measurement values. In this paper, the adaptive fading extended Kalman filter is used to fuse the position and attitude information of Kinect and IMU outputs, eliminate the noise, and thus improve the stability and positioning accuracy of the system. The integrated navigation system and filter structure are shown in Figure 4. Before the experiment, the parameters of Kinect and IMU are calibrated under the Ubuntu16.04 system.

## 7. Time Synchronization

To obtain the correct pose information of the robot by the Kalman filter, it is necessary to synchronize the output data of the multi-sensors in time. The output data by Kinect in this paper must be synchronized with the data collected by the MTi inertial sensor in time. Kinect and MTi work at different sampling frequencies. The highest sampling rate of Kinect is 30 FPS (frames per second). Due to the software mechanism and the calculation requirements between the systems, the Kinect frequency is set at 1 Hz. The sampling frequency of MTi can be set to 100 Hz, 50 Hz, or 25 Hz, and the actual navigation experiment is set at 100 Hz. For the problem of time synchronization, previous studies [27,28] have proposed the use of hardware synchronization, but this article does not use the hardware synchronization method. Instead, the RGB and depth images acquired by Kinect are saved with the Windows time by the program algorithm on the computers, and the data time tag set by the MTi is the same. Both are related to the UTC time in the personal computer. After both Kinect and MTi data are time-stamped, they can be interpolated and aligned by the program algorithm, so that the Kalman filter can be used for the data fusion.

## 8. Experiments

### 8.1. SIFT Algorithm Verification Experiments

Under the Ubuntu 16.04 system and OpenCV3, two example studies were conducted. The two images were matched by the classic SIFT algorithm and the improved SIFT algorithm. The results after matching are shown in Figure 5 and Figure 6, in which the above images are the matching results of the classical SIFT algorithm, and the bottom images are the matching results of the improved SIFT algorithm.

It can be seen from Table 1 and Table 2 that the performance of the improved algorithm is higher than that of the original algorithm due to the feature point decreasing (the percentage is about 27% for the two examples). In terms of time consumption, the optimization ratio is about 18% for the two examples as the improved algorithm simplifies the feature descriptor and reduces the feature vectors from the original 128 dimensions to 64 dimensions. Therefore, it has a significant advantage in regards to time-consumption.

### 8.2. Kalman Filter Algorithm Simulation Experiment

This section compares the performance of the AFEKF and the EKF algorithms through simulation experiments and public dataset tests.

First, the performance of each algorithm was verified by the simulation experiments. Under the MATLAB R2014a platform, a simulation study was carried out based on the SLAM algorithm simulation platform published by Professor Tim Bailey at the University of Sydney. The simulated map used is shown in Figure 7. The size of the map is 200 × 200 m. The solid line indicates the connection between the robot’s preset navigation points. The “*” indicates the feature points in the environment map; there are 35 landmark points and all landmarks are static landmarks. It is prescribed that the robot starts from the coordinates (0, 0) and moves the predetermined line counterclockwise at a certain rate for a circle.

A single simulation experiment was performed on the EKF and AFEKF algorithms. In the figures, the robot is represented by a triangle, the solid line is the actual motion trajectory of the robot, the broken line indicates the robot trajectory estimated by the SLAM method, and the "+" indicates the position of the landmark point estimated by the SLAM method. It can be seen that the estimated result of AFEKF–SLAM is more accurate than the EKF–SLAM method. The results are shown in Figure 8 and Figure 9, respectively. The five control points are chosen for the error comparisons with the above two methods in Table 3 and Table 4.

It can be seen from the above figures and tables that the errors of the EKF algorithm are larger than those of the AFEKF algorithm. When applying the EKF algorithm, the average error in the *X*-axis direction is 7.6 m, and the average error in the *Y*-axis direction is 2.4 m. After applying the AFEKF algorithm, the average error in the *X*-axis direction is 1.8 m, and the average error in the *Y*-axis direction is 0.4 m in Table 4.

### 8.3. Combined Positioning Experiment

The mobile robot platform used in this experiment is mainly composed of a WX-DP203 wheeled robot mobile platform, high-performance notebook computer, Kinect vision sensor, and IMU. During the whole experiment, the robot moves at a speed between 0.2 m/s and 0.3 m/s. In the experiment, the sampling frequency of the depth camera is set to 1 Hz, and the frequency of the IMU is 100 Hz. In order to verify the trajectory accuracy under different experimental conditions, three experiments were performed in this paper, corresponding to three trajectories. The experiment was carried out at the large conference center of the Institute of Space Science and Technology of Qianhu campus at Nanchang University. Before the experimental test, the key points of the waypoint path are set, the two-dimensional coordinates of the waypoint points are measured, and the positioning effects based on these points are compared.

#### 8.3.1. Straight Line Experiment

The robot walks along a straight line for a distance.

It can be seen from Figure 10 and Table 5 that the trajectory of IMU/Kinect is closer to the control points. With Kinect, the maximum offset of the robot trajectory on the X axis is 0.09 m. With IMU/Kinect, the maximum offset of the robot trajectory on the X axis is 0.038 m. The average errors in the *X*-axis direction are 0.06 m with Kinect and 0.02 m with IMU/Kinect. The average errors in the *Y*-axis direction are 0.02 m with Kinect and 0.01 m with IMU/Kinect in Table 6.

#### 8.3.2. Elliptical Motion Experiment

The figure below shows the comparison of the positioning results of the two algorithms in the elliptical motion. The running path of the robot is an elliptical trajectory.

The robot starts moving from the starting point with the coordinates (0, 0) in the upper right corner of Figure 11. It can be seen that the robot with both algorithms can return to the starting point. The trajectory offset of the Kinect algorithm is larger than that obtained by the IMU/Kinect algorithm because the latter performs better than the former. The seven waypoints (control points) are chosen for the comparisons in Table 7 and Table 8. With Kinect, the minimum value of the robot’s trajectory on the *X*-axis (negative half-axis) is −4.3 m, and on the *Y*-axis, the coordinate is −7.1 m. With IMU/Kinect, the robot’s trajectory is the smallest on the *X*-axis, with a value of −3.5 m, and on the Y axis, it is −6.6 m in Table 7. The experiment shows that the trajectory with IMU/Kinect is closer to the actual trajectory. Compared to the coordinates of the reference control points, the average errors in the *X*-axis direction are 0.55 m with Kinect and 0.06 m with IMU/Kinect. The average errors in the *Y*-axis direction are 0.39 m with Kinect and 0.12 m with IMU/Kinect in Table 8.

#### 8.3.3. Polygon Trajectory

The trajectory of this experiment is a hexagon and a quadrangle. The mobile robot starts from (0, 0), completes a circle clockwise, and finally returns to the starting point.

It can be seen from the trajectories in Figure 12 and the coordinates of the control points in Table 9 that the trajectory of the Kinect algorithm alone has the offsets to the control points, and is finally unable to return to the origin, whereas the trajectory of the IMU/Kinect algorithm is relatively closer to the control points. The nine waypoints (control points) are chosen for the comparisons in Table 9 and Table 10. With Kinect, the offset of the trajectory at the *X*-axis is 0.0 m at the destination, and the offset at the *Y*-axis is 0.2 m. With IMU/Kinect, the offset of the robot trajectory is 0.0 m on the *X*-axis and 0.05 m on the *Y*-axis at the destination in Table 9. The average errors in the *X*-axis direction are 0.11 m with Kinect and 0.06 m with IMU/Kinect. The average errors in the *Y*-axis direction are 0.22 m with Kinect and 0.15 m with IMU/Kinect in Table 10.

## 9. Comparison

Compared to the article [8], our paper presented three new approaches: (1) we introduced the improved SIFT algorithm, which has not been mentioned by [8]; (2) the absolute orientation algorithm was used to calculate the robot trajectory; and (3) we presented the adaptive and fading extended Kalman filter for processing the visual and IMU data, which has also not been mentioned by [8].

In addition to the simulation calculations, we conducted three field tests, including Straight Line, Elliptical Motion Experiment, and Polygon Trajectory Experiment with the IMU/Kinect. Our result accuracy is at the level of about ten centimeters in those tests. For example, the average error in the *X*-axis direction is 0.06 m with IMU/Kinect for 9 Waypoints in the polygon experiment, and the average error in the *Y*-axis direction is 0.15 m with IMU/Kinect. The article [8] did not achieve such an accuracy or failed to show how accurate it was. 

In general, our paper proposed some novel algorithms and improved the accuracy of the performance with visual and inertial sensors. Our method outperformed [8].

## 10. Conclusions

This paper has three highlights. First, the classic SIFT algorithm was improved, and experiments were performed that verified that the classical SIFT algorithm is insufficient in terms of both accuracy and time. The improved SIFT algorithm has a higher accuracy and reduces the matching time. Second, the AFEKF algorithm was used to replace the EKF algorithm, and the simulation results show that the trajectory based on the AFEKF algorithm is closer to the real trajectory. Third, the adaptive and fading extended Kalman filter was used for processing the visual and IMU data in a loose combination to compensate for the drawback of the single vision sensor.

The results of the polygon trajectory experiment show that the average errors decrease from 0.11 m with Kinect to 0.06 m with IMU/Kinect in the *X*-axis direction and from 0.22 m with Kinect to 0.15 m with IMU/Kinect in the *Y*-axis direction. Other test results were similar. These results verify that the method proposed by this paper can improve the accuracy compared to the single sensor.

However, there are some shortcomings in this paper. For example, the number of experiments is relatively small and the trajectory is relatively short. Other sensors, such as lidar, etc., are superior in indoor positioning. In the next step, we will study multiple data sets and conduct experiments in the tight combination to further eliminate errors.

## Figures and Tables

**Figure 1 sensors-19-01773-f001:**
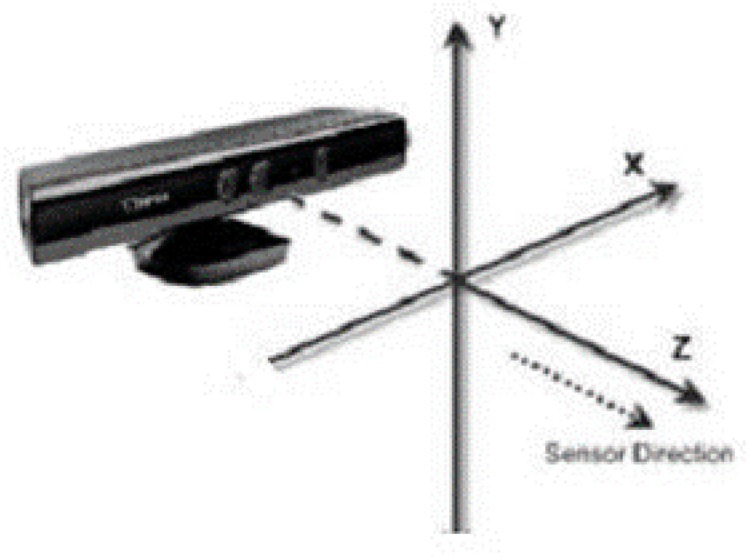
Kinect coordinate system.

**Figure 2 sensors-19-01773-f002:**
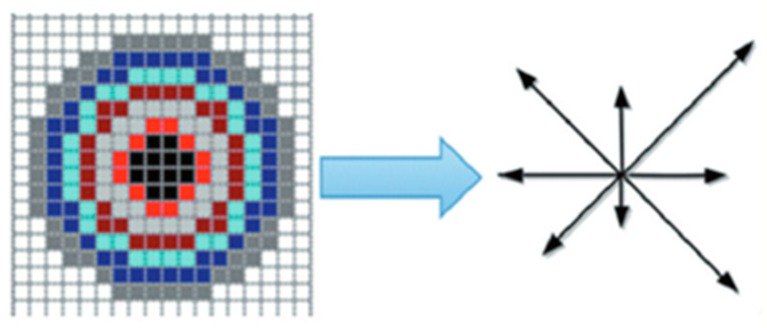
Feature description area.

**Figure 3 sensors-19-01773-f003:**
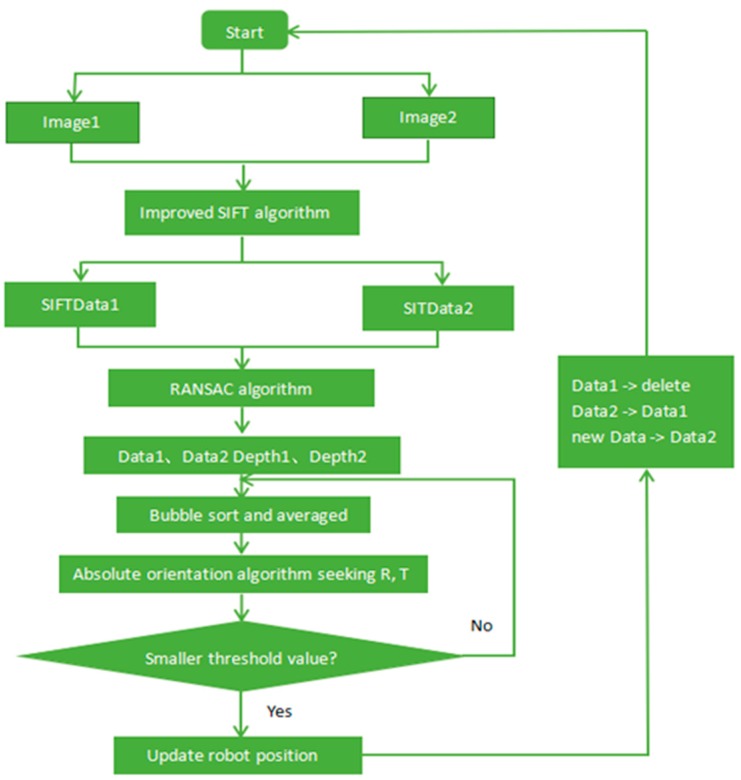
Flow chart based on visual positioning.

**Figure 4 sensors-19-01773-f004:**
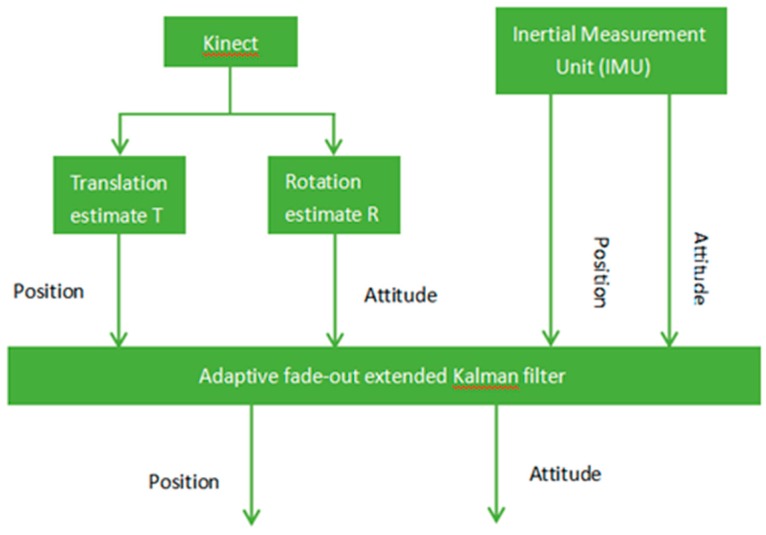
Loose combination and filtering system.

**Figure 5 sensors-19-01773-f005:**
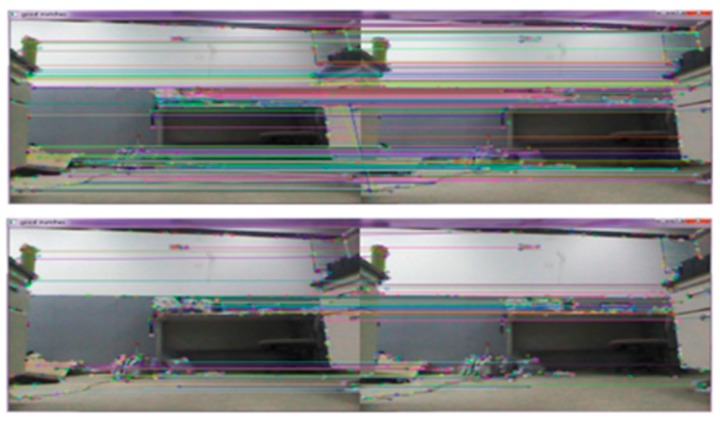
Comparison of matching results (example 1).

**Figure 6 sensors-19-01773-f006:**
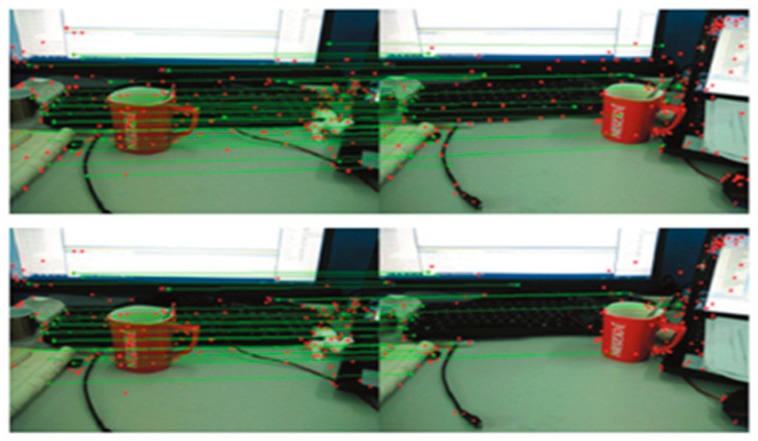
Comparison of matching results (example 2, the dataset of the images used in the experiment is from the Technical University of Munich, Germany).

**Figure 7 sensors-19-01773-f007:**
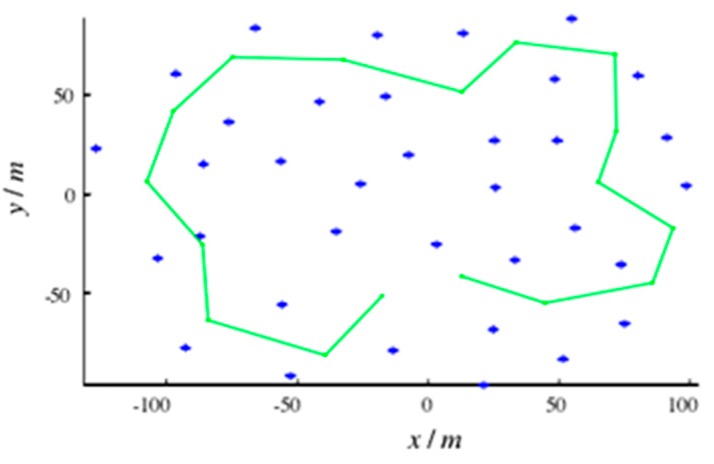
Simulation map.

**Figure 8 sensors-19-01773-f008:**
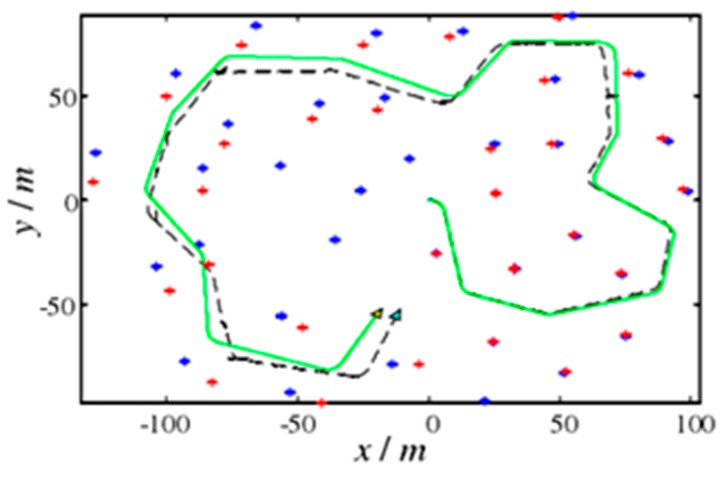
EKF algorithm simulation.

**Figure 9 sensors-19-01773-f009:**
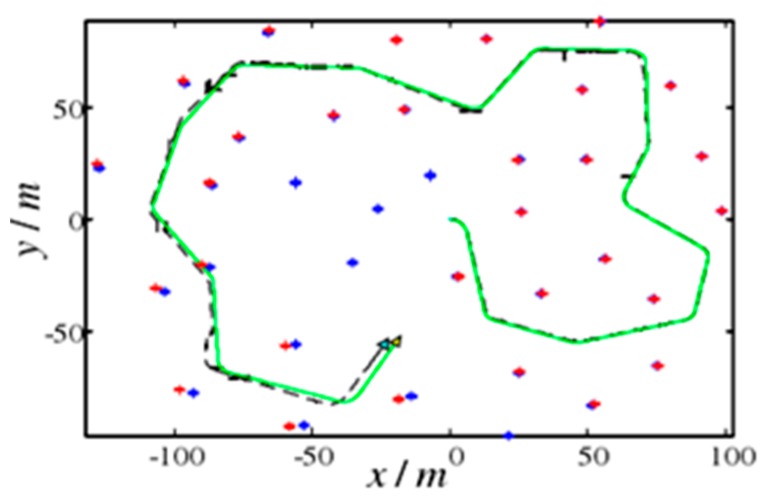
AFEKF algorithm simulation.

**Figure 10 sensors-19-01773-f010:**
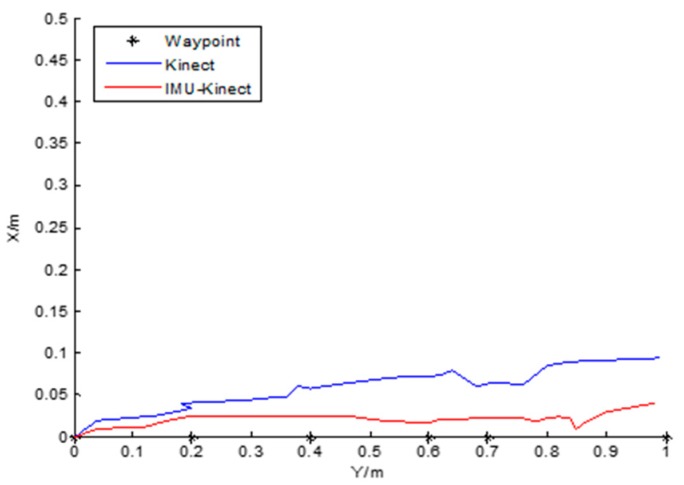
Straight track.

**Figure 11 sensors-19-01773-f011:**
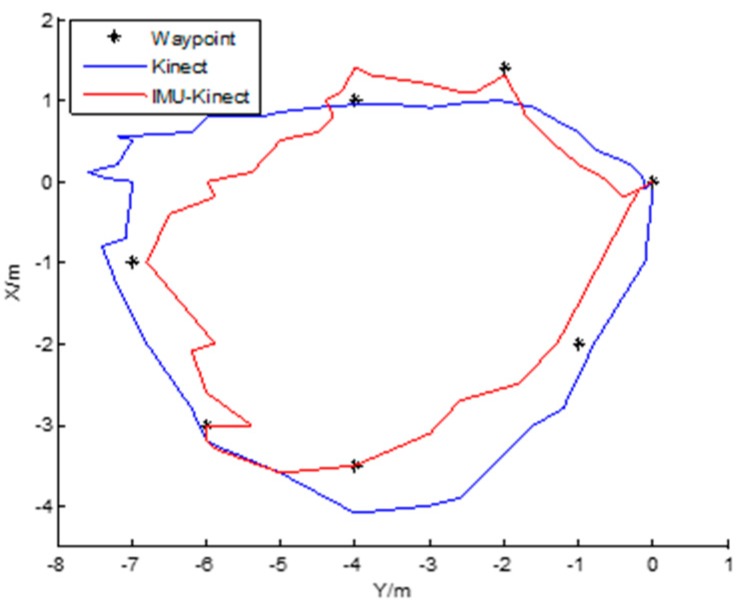
Elliptical track.

**Figure 12 sensors-19-01773-f012:**
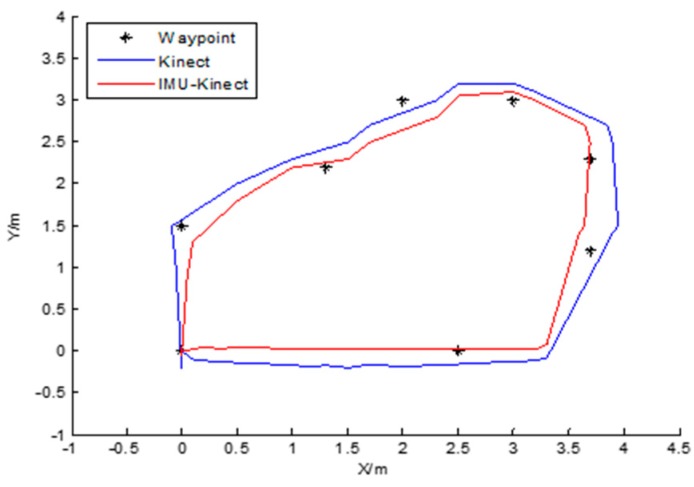
Polygon track.

**Table 1 sensors-19-01773-t001:** Comparison of matching result (example 1).

Percentage	SIFT Algorithm	Improved SIFT Algorithm	Optimization
Feature points	158	116	26.58%
time (s)	1.273	1.036	18.62%

**Table 2 sensors-19-01773-t002:** Comparison of matching result (example 2).

Percentage	SIFT Algorithm	Improved SIFT Algorithm	Optimization
Feature points	114	83	27.20%
time (s)	0.943	0.775	17.82%

**Table 3 sensors-19-01773-t003:** Optional five control points (in meters).

Number	1	2	3	4	5
Control Points	(0, −25)	(75, 60)	(18, 35)	(−93, −80)	(−20, −80)
EKF-SLAM	(0, −25)	(71, 60)	(10, 33)	(−82, −90)	(−5, −80)
AFEKF-SLAM	(0, −25)	(75, 60)	(18, 35)	(−100, −79)	(−22, −81)

**Table 4 sensors-19-01773-t004:** Error values of the points related to Table 3 (in meters).

Number	1	2	3	4	5	Average
EKF-SLAM(X)	0	4	8	11	15	7.6
EKF-SLAM(Y)	0	0	2	10	0	2.4
AFEKF-SLAM(X)	0	0	0	7	2	1.8
AFEKF-SLAM(Y)	0	0	0	1	1	0.4

**Table 5 sensors-19-01773-t005:** Control points’ coordinates in the straight line experiment (in meters).

Waypoints	1	2	3	4	5	6
True Values	(0, 0)	(0, 0.2)	(0, 0.4)	(0, 0.6)	(0, 0.7)	(0, 1)
Kinect	(0, 0)	(0.04, 0.18)	(0.06, 0.38)	(0.08, 0.64)	(0.062, 0.72)	(0.09, 0.98)
IMU/Kinect	(0, 0)	(0.024, 0.19)	(0.021, 0.4)	(0.015, 0.58)	(0.021, 0.69)	(0.038, 1)

**Table 6 sensors-19-01773-t006:** Absolute values of the coordinate errors (in meters).

	Kinect(x)	Kinect(y)	IMU/Kinect(x)	IMU/Kinect(y)
Average	0.06	0.02	0.02	0.01

**Table 7 sensors-19-01773-t007:** Control points’ coordinates of the elliptical motion experiment (in meters).

Waypoints	1	2	3	4	5	6	7
True Values	(0, 0)	(−2, −1)	(−3.5, −4)	(−3, −6)	(−1, −7)	(1, −4)	(1.4, −2)
Kinect	(0, 0)	(−2, −0.8)	(−4.3, −4)	(−3.3, −5.8)	(−0.7, −7.1)	(0.92, −4.2)	(−0.1, 0)
IMU/Kinect	(0, 0)	(−1.7, −1)	(−3.5, −4)	(−3, −6)	(−1, −6.6)	(1, −4.4)	(1.3, −2)

**Table 8 sensors-19-01773-t008:** Absolute values of the coordinate errors (in meters).

	Kinect(x)	Kinect(y)	IMU/Kinect(x)	IMU/Kinect(y)
Average	0.55	0.39	0.06	0.12

**Table 9 sensors-19-01773-t009:** Control points’ coordinates of the Polygon test (in meters).

Waypoints	1	2	3	4	5	6	7	8	9
True Values	(0, 0)	(2.5, 0)	(3.7, 1)	(3.7, 2.3)	(3, 3)	(2, 3)	(1.3, 2.2)	(0, 1.5)	(0, 0)
Kinect	(0, 0)	(2.5, −0.16)	(3.9, 1.5)	(3.85, 2.7)	(3.1, 3.2)	(2, 2.9)	(1.3, 2.5)	(−0.1, 1.4)	(0, −0.2)
IMU/Kinect	(0, 0)	(2.5, 0.016)	(3.3, 1.5)	(3.7, 2.3)	(2.9, 3.1)	(2, 2.6)	(1.3, 2.3)	(0.1, 1.3)	(0, −0.05)

**Table 10 sensors-19-01773-t010:** Absolute values of the coordinate errors (in meters).

	Kinect(x)	Kinect(y)	IMU/Kinect(x)	IMU/Kinect(y)
Average	0.11	0.22	0.06	0.15

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
