# Peer review of "Mobile Robot Indoor Positioning Based on a Combination of Visual and Inertial Sensors"

_sensors, 2019, doi:10.3390/s19081773_

Round 1
Reviewer 1 Report
in general: ask a native speaker to check your text, some sentences are not understandable, e.g. "Obtain scale values and improve feature points for scale changes resistance, and finally extract feature descriptors" or you are using wrong words, e.g. "deep data stream"
in general: check for relevant references - there is much more technology in the world defining current benchmarks
82: "fx and fy are the internal parameters of the RGB camera" - which one?
104: "The larger the scale parameter, the more the feature of the image is left" - what???
161: good idea about applying circular templates, but describe it in a better way! Maybe a figure helps
195: Phi, Omega, Kappa are not shown in equation (10)
183: rewrite the entire chapter, it is not readable
183: isn't it a relative orientation insteat an absolute one?
213: where do you apply RANSAC?
225: "infrared rays have the property of waves" - yes, all waves have
231: "Using the idea of the sequential iteration, …", the entire sentence is not readable
257: chapter seems to be copied from a textbook, highlight your contributions otherwise shorten it massively
257: don't you calibrate your IMU?
292: chapter seems to be copied from a textbook, highlight your contributions otherwise shorten it massively
344: where and how did you use the Kalman filter? Did you use it as a black box and just feed IMU and Kinect data in?
344: how do you ensure time synchronization between IMU and Kinect data?
372: one exaple is not enough to claim methat A is better than method B
422: "is more accurate” – provide quantitative results
Author Response
Thank you very much for your opinions. We have attached the response to you for your reference.

Reviewer 2 Report
The authors present a positioning system that fuses the information from an improved SIFT algorithm that uses depth information for translation and rotation estimation, and the measurements from an inertial measurement unit (IMU).
The article is well written, but there are some issues the authors should acknowledge and resolve:
- The authors should add what IMU stands for in the abstract since it is not explained previously.
- The authors should use the same colors for Kinect and IMU-Kinect for Figures 7 as what they used for Figures 8 and 9, to avoid confusion.
- Additionally to the variance in percentage, the authors should provide the distance between the true values and the Kinect and IMU-Kinect estimations for each way-point greater than 1 in Table 2, 3, and 4, as a measurement for error. Additionally, the tables should show an average of such an error.
- In the same manner, the authors should provide a similar table to present the values seen in Figure 5 and Figure 6 to compare the performances between EKF and AFEKF.
- The error of the proposed system, depending on the experiment, is between 5 and 7.6 % variance. How does this compare with other positioning systems?
- Biggest issue: the technique the authors propose is very similar to the work presented in:
V. Sazdovski and P. M. G. Silson, "Inertial Navigation Aided by Vision-Based Simultaneous Localization and Mapping," in IEEE Sensors Journal, vol. 11, no. 8, pp. 1646-1656, Aug. 2011.
doi: 10.1109/JSEN.2010.2103555
The authors need to acknowledge this work and present how is their work different and/or how it outperforms it.
Author Response

(The authors gave the same response as above.)

Round 2
Reviewer 1 Report
Dear authors,
you made a lot of Progress. Congratulations! But nevertheless, I would highly recommend to ask a native speaker to read and correct your paper. There are a lot of phrases sounding not correct!
Regards
Author Response
Dear Reviewer,
Thank you very much for your time and efforts on us.
Best regards,
Hang Guo, Ph.D.

Reviewer 2 Report
The authors have acknowledged and fixed most of the issues this reviewer raised in the last revision iteration. However, even though they have added the work presented in:
V. Sazdovski and P. M. G. Silson, "Inertial Navigation Aided by Vision-Based Simultaneous Localization and Mapping," in IEEE Sensors Journal, vol. 11, no. 8, pp. 1646-1656, Aug. 2011.
doi: 10.1109/JSEN.2010.2103555
They failed to establish the differences between that work and theirs; the authors only mentioned in the introduction that it is similar to their work. However, if it is similar, then what is the contribution of this paper?
The authors need to establish how is their methodology different to that work; how is it better? If this is not asserted, there is no reason for this work to merit publication.
Author Response

(The authors gave the same response as above.)
